# Differential Reinforcement without Extinction: An Assessment of Sensitivity to and Effects of Reinforcer Parameter Manipulations

**DOI:** 10.3390/bs14070546

**Published:** 2024-06-28

**Authors:** Hannah MacNaul, Catia Cividini-Motta, Kayla Randall

**Affiliations:** 1Department of Educational Psychology, University of Texas at San Antonio, San Antonio, TX 78249, USA; 2Department of Child and Family Studies, University of South Florida, Tampa, FL 33620, USA; catiac@usf.edu; 3Department of Psychology, Georgia Southern University, Statesboro, GA 30460, USA; krandall@georgiasouthern.edu

**Keywords:** differential reinforcement, extinction, functional communication, reinforcer parameters

## Abstract

Although functional communication training (FCT) usually includes extinction, withholding reinforcement is not possible or ethical with certain individuals, for some topographies of problem behavior, or in certain contexts. The current study evaluates the effects of two variations of FCT, both without extinction, on problem behavior and communication. Further, the intervention procedures were designed to evaluate participant reactivity to reinforcer parameters (e.g., magnitude, delay, and quality) in the context of the FCT variations. The parameter sensitivity assessments were effective at identifying relevant reinforcer parameters for each participant and both FCT interventions were effective in decreasing problem behavior and increasing communication for all participants. The results demonstrated that FCT was effective regardless of which reinforcer parameter was manipulated. Moreover, all sessions were conducted in participants’ homes and caregivers reported high degrees of social validity for the intervention procedures and outcomes.

## 1. Introduction

Functional communication training (FCT) is a commonly prescribed intervention to decrease problem behavior (e.g., aggression, elopement) and increase appropriate communication [1,2]. In this intervention, reinforcement is delivered for the functional communication response (FCR) and is either withheld or degraded for problem behavior. Extinction, which entails withholding reinforcement following the occurrence of a previously reinforced response, is a supplemental and frequently used procedure in research and clinical practice evaluating FCT. 

Previous research suggests that FCT interventions that incorporate extinction for problem behavior are more effective compared to those that do not include extinction [3,4]. For instance, Hagopian et al. [4] evaluated the efficacy of FCT with and without various treatment components in a large sample of individuals with problem behavior. A total of 11 out of 21 cases experienced FCT without extinction in which the consequence for problem behavior and the FCR were the same. Not only was FCT without extinction ineffective at reducing problem behavior by 90% from the baseline for all cases, but overall, problem behavior increased an average of 17.4% from the baseline. As a result, the remainder of cases included extinction, punishment, or a combination of the two to reduce problem behavior by 90% of baseline levels. While these may be effective, previous literature has documented the pitfalls of extinction (e.g., extinction burst, extinction-induced aggression; [5,6]), and the Behavior Analyst Certification Board (BACB^®^) recommends utilizing reinforcement-based approaches before considering punishment procedures [7]. Additionally, autistic individuals who have reported experiencing extinction have expressed concerns regarding this procedure, leading to calls for future research to explicitly evaluate the relative efficacy and efficiency of interventions that include and do not include extinction and participants’ preference for these interventions [8].

Given FCT without extinction with equated consequences for problem behavior and appropriate communication may be ineffective at decreasing problem behavior [4], a small but growing body of research has employed reinforcer parameter manipulations when implementing FCT (or the differential reinforcement of alternative behavior) without extinction and attained therapeutic effects. In these studies, reinforcer parameters—quality, immediacy, and magnitude—were systematically manipulated such that reinforcement contingencies favored alternative responding (e.g., greater reinforcement is provided for the FCR) and were less favorable towards problem behavior. Notably, Athens and Vollmer [9] manipulated either one parameter or a combination of reinforcer parameters to evaluate optimal decreases in problem behavior. Generally, the greatest reduction occurred when a combination of quality, magnitude, and immediacy of reinforcement was manipulated to favor the FCR. This approach has been applied in a handful of studies to treat problem behavior [10,11,12,13] and summarized in a recent review of the literature [14]. However, in these studies, the authors evaluated the impact of various parameter manipulations on problem behavior and the FCR, resulting in lengthy treatment evaluations. One way to address this limitation could be to conduct preassessments to identify optimal reinforcer sensitivities to be manipulated in the treatment evaluation [15,16,17].

Kunnavatana et al. [17] investigated such a novel approach by evaluating parameters of reinforcement in isolation and relative to one another based on a participant’s emission of an arbitrary response (e.g., switch touching). Specifically, researchers evaluated sensitivity by measuring response allocation to two concurrently available response options, each associated with a differing reinforcer parameter (e.g., large magnitude versus immediate access). Generally, the results indicated that participants engaged in differentiated responding when exposed to various parameters allowing the researchers to identify, prior to implementing the treatment evaluation, the most-sensitive and least-sensitive reinforcer parameter for each participant. Although some researchers have suggested a translational model for identifying reinforcer sensitivities may be helpful [16], this approach may prove unfeasible for individuals who engage in problem behavior as they may be reluctant to sit at a computer for extended periods of time to complete the assessment. Thus, more research on the identification of sensitivity to reinforcer parameters for children with problem behavior is needed, with special consideration of the environments for which these preassessments are conducted. To our knowledge, no studies have completed these types of assessment in natural settings (e.g., home). 

Taken together, studies have shown that the identification of reinforcer sensitivity is possible and useful in a preassessment [16], problem behavior can decrease as a function of reinforcer parameter manipulations [9], and sensitivities to parameters vary by participants [17]. Thus, the purpose of this study was to replicate previous research [17] and to extend the literature in a few key areas: (a) inclusion of stimulus magnitude and delay value sensitivity assessments to identify optimal magnitude and delay values for each participant’s contingencies, (b) evaluate the efficacy and efficiency of two variations of FCT without extinction, designed according to the results of the relative parameter sensitivity assessments, (c) evaluate the efficacy of this procedure in a natural setting (i.e., home), and (d) assess social validity on behalf of the caregivers for the study procedures employed.

## 2. Method

### 2.1. Participants and Settings

Participants included four children with ASD: Roberto, Dominick, Maggie, and Leo. According to their caregivers, all participants engaged in problem behavior that was reported to hinder their academic and/or daily functioning. Participants were recruited for this study via flyers posted around the community; therefore, the severity or intensity of participant’s problem behavior varied. Roberto was a four-year-old Latinx male who was able to communicate vocally using 1–2 word phrases. The primary language spoken in the home was Spanish, but Roberto also communicated vocally using a few English words and phrases. Additionally, his mother preferred that English was spoken during sessions and that the FCT intervention used communication responses in English. Dominick was a two-year-old Caucasian male and English was the primary language spoken in the home. At the time of the study, Dominick had not received intervention services and communicated using gestures (e.g., pointing) and physical guidance (e.g., physically prompting his caregiver to obtain items or go to a desired area). Maggie was a two-year-old Caucasian female and English was the primary language spoken in the home. She also had no history of intervention services prior to her enrolling in this study and she could communicate vocally using 2–3-word phrases. Leo was a 10-year-old Latinx male and English was the primary language spoken in the home. Leo’s mother also frequently spoke Spanish, but preferred that communication responses in the intervention be implemented in English. Leo could communicate vocally with his caregivers using one-word vocal approximations, but these vocalizations were not understood by most individuals (e.g., primary investigator, teacher). All sessions were conducted in the participant’s home. Specifically, sessions took place in the living room for Roberto and Maggie, the bedroom for Dominick, and in an outside enclosed area for Leo. All procedures were conducted by the researcher, but caregivers were always present in the home and often informally observed sessions. Materials included a paper and pen, cell phone with the data collection application, Countee©, and stimuli used as reinforcers. A video camera was also present as sessions were recorded for data collection purposes.

### 2.2. Response Definitions and Measurement

Data were collected on three dependent measures: stimulus selection, problem behavior, and appropriate communication. Stimulus selection consisted of the participant pointing to or making physical contact with one of the available stimuli and is reported as the percentage of opportunities. The operational definition of problem behavior was individualized for each participant. Appropriate communication was conceptualized as an FCR for each participant. All target FCRs were selected based on preference from caregivers about the mode (e.g., vocal or pictorial response) and language (e.g., English or Spanish) and are listed in Table 1. Given that the duration of the sessions during the treatment evaluation varied, problem behavior and appropriate communication are reported as the rate, which was calculated by dividing the number of occurrences by the total duration of the session in minutes.

### 2.3. Interobserver Agreement (IOA) and Treatment Integrity

IOA was calculated for an average of 47.5% (range, 25–100%) of sessions across participants. The mean IOA score was 98.3% (range, 87.9–100%), 99.5% (range, 98.3–100%), 97.2% (range, 90–100%), and 99.6% (range, 98.5–100%) for Roberto, Dominick, Maggie, and Leo, respectively. Treatment integrity was assessed for an average of 43.6% (range, 25–100%) of sessions across all participants. The average treatment integrity score for each participant was 99.5% (range, 98.3–100%), 98.4% (range, 94–100%), 99.7% (range, 94.3–100%), and 99.8% (range, 98.8–100%) for Roberto, Dominick, Maggie, and Leo, respectively. 

### 2.4. General Procedure

During the functional analysis (FA), we employed a multielement experimental design. A reversal design was employed during the stimulus magnitude assessment, the delay value assessment, and the relative parameter sensitivity assessments. To note, the replication of results in the reversal conditions of the stimulus magnitude and delay value sensitivity assessments are at the specific dimension of each reinforcer rather than the titration procedure used to arrive at the terminal values. For the FCT evaluation, a multiple-baseline design across participants was used for Roberto and Dominick, while a reversal design was used for Maggie and Leo. We counterbalanced the sequence of conditions across participants and Roberto experienced the conditions twice to assess for replication of the effects within a participant. Figure 1 displays the experimental operations flow chart which outlines each step in the experimental process and lists the goal of each operation. Then, detailed descriptions of each experimental operation are provided in their respective sections (e.g., Section 2.9, Section 2.10, Section 2.11 and Section 2.12 describe the FCT evaluation).

### 2.5. Preference and Reinforcer Assessments

A paired-stimulus preference assessment (PSPA) [18] and a concurrent-operant reinforcer assessment [19] were conducted with all participants to identify higher- and lower-quality reinforcers to be used in the quality manipulations. Stimuli were classified as higher or lower quality if they were chosen in 70% or more of opportunities (higher-quality) or less than 30% of opportunities but at least once during the preference assessment (lower-quality), respectively (see Table 1). For problem behavior maintained by social negative reinforcement in the form of escape (i.e., Leo), the quality contingency included either an enriched break or an unenriched break. The enriched break consisted of a break plus access to highly preferred items [20], and an unenriched break consisted of a break only. The reinforcer magnitude was adjusted based on the duration of the break.

### 2.6. Functional Analysis

A functional behavior assessment consisting of an indirect assessment (Functional Analysis Screening Tool; [21]) and an FA [22] were conducted with each participant to identify the function(s) of each participant’s problem behavior. Based on results of the FA, the function of Roberto, Dominick, and Maggie’s problem behavior was access to tangibles and Leo engaged in problem behavior to escape from academic demands (e.g., reading a book). Data and graphs are available upon request.

### 2.7. Stimulus Magnitude and Delay Value Assessments

The purposes of these assessments were to determine if participants were sensitive to each parameter in isolation and to identify specific values associated with exclusive responding toward one of the response options. For both magnitude and delay value assessments, the procedures resembled a PSPA [18]. At the start of each session, 2 exposure trials were conducted, followed by 10 choice trials. During each choice trial, two picture cards were presented to the participant. These pictures cards contained the target stimulus (e.g., iPad^®^) which was the higher-quality item from the preference assessment. 

For the stimulus magnitude assessment, the first session (Phase A) included presenting the participant with identical pictures (i.e., same size, outlined in white) of the target stimulus, and the selection of either picture resulted in immediate access to the reinforcer at the same magnitude value. Consistent with previous research [23], the initial magnitude value was set to 5 s for all participants. However, for Dominick, the initial value was changed to 10 s after the first two trials of session one because after experiencing the consequence of 5 s a couple of times, Dominick stopped responding in subsequent trials. We hypothesize that 5 s access to the reinforcer was not enough to maintain his responding and therefore, the initial value was set to 10 s for Dominick only. For the following sessions (Phase B), two cards were presented to the participant, but the card associated with the larger-magnitude reinforcer was now displayed with a larger card compared to the card associated with the smaller-magnitude reinforcer. The size of the large card remained the same although the magnitude values increased across sessions. The selection of the larger card resulted in a greater magnitude of that reinforcer, whereas selection of the original card (i.e., same size as in session one) resulted in the same reinforcer but at the original magnitude (e.g., 5 s). Across sessions, the magnitude of the reinforcer increased in increments of 5 s, until the participant exclusively allocated their behavior to the larger-magnitude reinforcer for all 10 trials. During the subsequent session we reversed to the initial magnitude values (i.e., 5 s and 5 s; Phase A) and then returned to the terminal values from the initial Phase B. These values were used as the high- and low-magnitude values for the remaining phases of the study. Data on problem behavior were collected throughout the entirety of the session. 

The delay value assessment employed the same arrangement as the stimulus magnitude assessment, but instead of changing the magnitude of a reinforcer, we increased the delay value associated with one response option. Additionally, the card associated with delayed delivery had a red border while the card associated with immediate reinforcer delivery had a green border. While the delay value changed, the magnitude was held constant and it consisted of the terminal value identified in the stimulus magnitude assessment (i.e., 20 s for Roberto, 35 s for Dominick, 15 s for Maggie, and 30 s for Leo). Similar to the stimulus magnitude assessment, the immediate consequence remained at a 0 s delay, but the delayed consequence increased in increments of 5 s. The same reversal design was implemented, and the final delay value was used as the delayed (D) reinforcer for the remaining phases of the study.

### 2.8. Relative Parameter Sensitivity Assessments 

This assessment directly replicated procedures from Kunnavatana et al. [17] and was used to determine if the participant’s behavior was sensitive to a specific reinforcer parameter when both were available simultaneously. Across all parameter sensitivity assessments, the response stimulus was a touch light covered with different colors of tissue paper and an index card of the same color under the touch light. Two touch lights, each associated with one of the consequences available (i.e., larger magnitude reinforcer; immediate reinforcer), were available in each session. See Table 2 for the specific parametric values implemented for each participant. 

For all relative parameter sensitivity assessments, sessions included 10 trials and at least three sessions were completed per phase. To start (Phase A), the researcher presented two touch lights to the participant, each associated with a different contingency. Contingent on stimulus selection, the corresponding consequence was delivered. Placement of the lights was switched from left to right across every trial. In the quality versus magnitude assessment, response allocation was measured between a higher-quality, low-magnitude reinforcer and a lower-quality, high-magnitude reinforcer. In the quality versus immediacy assessment, response allocation was measured between a lower-quality, immediately available reinforcer and higher-quality, delayed reinforcer. In the magnitude versus immediacy assessment, we measured response allocation between a low-magnitude, immediately available reinforcer and a high-magnitude, delayed reinforcer. Phase B served as a tracking test in which the same contingencies were in effect, but the stimuli associated with each consequence was switched. This was performed to ensure that the participant was tracking the contingencies and not making their selection based on other factors such as color. 

We conceptualized the most-sensitive reinforcer parameters as the combination for which participants selected response options for those parameters most often. Conversely, we conceptualized the least-sensitive reinforcer parameters as those for which participants selected the least often. We used these varying sensitivities to program two conditions of FCT, least sensitive and most sensitive. 

### 2.9. Treatment Evaluation: DRA without Extinction

The intervention portion of the study was completed to evaluate the effects of two iterations of FCT without extinction on levels of problem behavior and appropriate communication (i.e., FCR). Given the FCR was not in the participants’ repertoire prior to their involvement in this study, we conducted mand training prior to the baseline phase. During mand training, independent and prompted FCRs were continuously reinforced using most-to-least prompting, and all participants acquired the FCR in three or four sessions. 

During the FCT evaluation, each session lasted until 10 reinforcers were delivered. Therefore, session durations varied across participants and conditions, but on average sessions lasted less than 10 min with the longest session being 25 min (Session 13 for Dominick). At the start of each session, the researcher presented the specified evocative situation to the participant. This included removing access to the tangible reinforcer but keeping them in sight for Roberto, Maggie, and Dominick and prompting Leo to read from a book. Consequences comprised of differing parameters based on the results of the sensitivity assessments were provided for problem behavior and FCRs on a fixed ratio (FR) 1 schedule. The parameters of reinforcement during treatment were not signaled (i.e., larger card, outlined in red/green) in the way they were during the stimulus magnitude and delay value assessments; however, at the start of each condition, the participant was prompted to engage in the FCR in order to contact the contingency for the alternative response. The prompted FCR and the corresponding reinforcing consequence did not count toward the 10 total reinforcers delivered during each session. 

### 2.10. Baseline

Consistent with previous research [9,17], both problem behavior and the FCR were reinforced in the baseline. However, in our study, reinforcement contingencies favored problem behavior. This modification was performed to ensure we would have an opportunity to evaluate the different parameter manipulations and minimize the likelihood that the FCR would extinguish. For example, if the participant was most sensitive to immediacy and least sensitive to magnitude (i.e., Dominick), during baseline, problem behavior resulted in an immediate, high-magnitude reinforcer, whereas the FCR resulted in a delayed, low-magnitude reinforcer. 

### 2.11. FCT Least Sensitive (FCT-LS)

During FCT-LS, we manipulated the least-sensitive reinforcer parameter with contingencies favoring the FCR while keeping the most-sensitive parameter constant. For instance, during FCT-LS for Dominick, the immediacy remained constant as both responses resulted in delayed access to the preferred item (40 s delay); however, the FCR produced access to the preferred item for 35 s (high magnitude; HM) while problem behavior produced access to the item for 10 s (low magnitude; LM). 

### 2.12. FCT Most Sensitive (FCT-MS)

During FCT-MS, we manipulated the most-sensitive reinforcer parameter with contingencies favoring the FCR while keeping the least-sensitive parameter constant. For instance, during FCT-MS for Dominick, the magnitude remained constant as both responses resulted in 10 s access to the reinforcer; however, the FCR was reinforced immediately while problem behavior resulted in a 40 s delay. Additionally, if the contingency for either problem behavior or FCR included a delay, we reset the delay interval contingent on repeated emission of the first response topography. For example, if problem behavior was the first response emitted and a delayed consequence was programmed, each additional occurrence resulted in resetting the delay interval. If the alternative response (i.e., FCR) was emitted during the delay interval and had a different consequence, then that consequence was delivered. All participants’ contingencies for the FCT evaluation are depicted in Table 3. 

### 2.13. Social Validity Measure

Following the treatment evaluation, caregivers were shown brief 30 s videos of the first baseline session and the last session of both FCT conditions and asked to complete a social validity questionnaire (copy available as Appendix A). The questionnaire consisted of statements regarding the effectiveness and acceptability of the intervention. All responses were confidential and unidentifiable as the caregiver was instructed to mail the completed form to the researcher. 

## 3. Results

Figure 2, Figure 3, Figure 4, Figure 5, Figure 6 and Figure 7 contain the results for all participants. Figure 2 contains the results of the stimulus magnitude assessment for Roberto, Dominick, Maggie, and Leo. Throughout the assessment, problem behavior did not occur. For all participants, when both contingencies resulted in access to the functional reinforcer at the same magnitude (Phase A), responding was variable, with an approximate equal distribution of responses toward each available option. As the magnitude-value of one of the cards increased (e.g., 5, 10, 15 s), participants began selecting the card associated with the larger-magnitude reinforcer, ultimately leading to a session with 100% allocation to the larger magnitude reinforcer. 

Eventually, all participants responded exclusively to the larger-magnitude response option. This occurred when the high-magnitude values were 20, 15, 35, and 30 s for Roberto, Dominick, Maggie, and Leo, respectively. Additionally, the response during the second Phase A when the same initial magnitude values (i.e., 5 s and 5 s) were in effect and in the second Phase B, when the terminal differing magnitude values (i.e., 5 s and 20 s) were in effect, was similar across all participants. 

The results of the delay value sensitivity assessments are depicted in Figure 3. Initially, when both response options corresponded to immediate delivery of the functional reinforcer at the same magnitude (i.e., large magnitude value), the response was variable with an approximate equal distribution of responses toward each available option. As the delay associated with one of the options increased (e.g., 5, 10, 15 s), participants began selecting the response option associated with immediate reinforcer delivery (e.g., 0 s delay). All participants eventually allocated 100% of their responses to the option associated with immediate reinforcer delivery in nine sessions or less. This occurred when the delay values were 25, 40, 15, and 35 s for Roberto, Dominick, Maggie, and Leo, respectively. Only Dominick engaged in problem behavior during this assessment; therefore, at session seven, a rule was introduced, “Green card, no wait”. This rule was stated by the researcher at the start of each session for the remainder of Dominick’s assessment. Ultimately, this reduced problem behavior and resulted in exclusive allocation to the immediate reinforcer contingency. 

The results of the relative parameter sensitivity assessment are depicted in Figure 4 and Figure 5, and the contingencies associated with each assessment are reported in Table 2. Roberto and Dominick’s relative parameter sensitivity assessments are displayed in Figure 4. Roberto’s behavior was more sensitive to quality compared to magnitude (top left panel), quality compared to immediacy (middle left panel), and immediacy compared to magnitude (lower left panel). Problem behavior was low across these assessments except for the immediacy versus magnitude assessment, and problem behavior corresponded to sessions in which the delayed response option was chosen. Based on the results of these assessments, we concluded that Roberto’s behavior is most sensitive to the quality of reinforcer and least sensitive to the magnitude of a reinforcer. 

Dominick’s relative parameter sensitivity assessment results are depicted on the right side of Figure 4. Dominick’s behavior was more sensitive to quality compared to magnitude (top right panel), immediacy compared to quality (middle right panel), and immediacy compared to magnitude (bottom right panel). Dominick emitted few instances of problem behavior throughout most of the relative parameter sensitivity assessments; however, similar to Roberto, he engaged in elevated rates of problem behavior when the delay option was chosen during the delay sensitivity assessments (middle and bottom panel). Therefore, we concluded that Dominick’s behavior is most sensitive to the immediacy of reinforcer delivery and is least sensitive to the magnitude of a reinforcer. 

Maggie’s behavior (Figure 5) was more sensitive to quality compared to magnitude (top left panel), quality compared to immediacy (middle left panel), and magnitude compared to immediacy (bottom left panel). Problem behavior generally remained low throughout these assessments, with one burst observed in session two of the quality versus magnitude assessment. These results suggest that Maggie’s behavior is most sensitive to the quality of reinforcer and least sensitive to the magnitude of a reinforcer. 

Leo’s results showed that his behavior (Figure 5) was more sensitive to quality compared to magnitude (top right panel), quality compared to immediacy (middle right panel), and magnitude compared to immediacy (bottom right panel). As with Roberto and Dominick, Leo also engaged in problem behavior when the delay option was chosen. These findings indicate that Leo’s behavior is most sensitive to the quality of the reinforcer and least sensitive to the immediacy of reinforcer delivery.

Figure 6 and Figure 7 display the results of the treatment evaluation, and Table 3 contains the contingencies in effect for each participant. The rate of problem behavior and FCRs was used to determine the relative efficacy of the FCT variations, whereas session duration served as a measure of efficiency. Roberto and Dominick (Figure 6) engaged in high levels of problem behavior and low levels of communication during baseline. In regard to problem behavior, both the FCT-MS and FCT-LS condition resulted in low levels compared to the baseline. Overall, FCRs increased compared to the baseline; however, fewer FCRs occurred in the FCT-LS condition compared to the FCT-MS condition. Additionally, sessions were shorter in the FCT-MS condition compared to the FCR-LS condition. 

The results of the treatment evaluation for Maggie (top panel) and Leo (bottom panel) are depicted in Figure 7. Overall, both FCT conditions were effective in reducing problem behavior and increasing communication; however, the levels of the FCR and session durations differed. For Maggie, the rates of FCRs were similar across both FCT conditions, whereas Leo emitted the FCR more often during the FCT-LS condition. Moreover, session durations were similar across FCT conditions for Maggie, but for Leo, sessions of the FCT-LS condition were shorter in duration compared to the FCT-MS condition. 

Social validity was assessed with caregivers at the completion of the study. The results yielded an average score of 4 (range, 4 to 4) on a scale from 1 = strongly disagree to 4 = strongly agree. Specifically, caregivers noted a decrease in their child’s challenging behavior, an increase in their communication, and reported the intervention to be appropriate for their child’s age and topography of problem behavior. They also indicated that they would be likely to recommend this intervention to others. Finally, all caregivers received training on the FCT intervention and reported using it in the home and community setting following the completion of the study.

## 4. Discussion

We used a stimulus magnitude and delay value assessment to determine optimal consequences (i.e., magnitude, delay, quality) for each participant. These values were used in a series of relative parameter sensitivity assessments completed to identify each participant’s most- and least-sensitive reinforcer parameter which were incorporated into one of two FCTs without extinction conditions (i.e., FCT-LS and FCT-MS). In the current study, both FCT conditions were effective in reducing problem behavior and increasing FCRs, indicating that the parameter sensitivity assessments were effective at identifying relevant reinforcer parameters for each participant but also demonstrating that FCT was effective at reducing problem behavior regardless of which parameter was manipulated. 

Our results provide new innovations relative to existing research on FCT without extinction. Specifically, our results are similar to previous research showing that, when reinforcer parameters are evaluated and subsequently manipulated to favor the alternative behavior, FCT without extinction is effective in reducing problem behavior and increasing alternative behavior [9,11,13,17]. Moreover, a rapid reduction in problem behavior was observed during the treatment evaluation, which is similar to the patterns of responding observed in Kunnavatana et al. [17]. Kunnavatana et al. hypothesized that the rapid reduction in problem behavior in the treatment evaluation was due to the strong history of reinforcement that was provided for the alternative response in the pretreatment assessment procedures. It is possible that repeated exposure to reinforcement contingent on any alternative response (e.g., stimulus selection, target touching, and emission of the FCR) during the preassessments and mand training resembled a serial-DRA procedure [24,25,26], resulting in immediate and robust treatment effects. However, this history of reinforcement did not lead to persistent occurrence of the FCR in the baseline which suggests that, similar to Hagopian et al. [4], equated consequences alone would not have resulted in therapeutic gains. Further investigation into the effects of these assessments on the outcomes of the subsequent treatment evaluation is needed. 

Another important finding related to the stimulus magnitude and delay value assessments is that this procedure allowed us to identify the optimal reinforcer magnitude needed to maintain behavior without using reinforcer magnitudes larger than necessary. These magnitudes were smaller than in previous research. For example, the largest magnitude and longest delay was 35 s and 40 s, respectively. Conversely, the largest magnitude values were 45 s and 180 s and the longest delay values were 60 s and 280 s in Athens & Vollmer [9] and Kunnavatana et al. [17], respectively. Although programming shorter reinforcer magnitudes to maintain low levels of problem behavior may seem paradoxical, previous research has demonstrated it may be advantageous to identify the smallest amount of a reinforcer necessary to maintain behavior [27]. Using the smallest magnitude necessary allows for more opportunities to contact reinforcement within a certain time frame (e.g., session) and may be more feasible and acceptable to deliver in natural environments. 

Our results differ from previous studies in a few ways. First, in our study, FCT without extinction was effective for all participants even when a single reinforcer parameter was manipulated. Conversely, Athens and Vollmer [9] had three participants (Justin, Lana, and Kenneth) for whom FCT without extinction was not effective when a single parameter was manipulated. Similarly, in Briggs et al. [11], when only the quality parameter was manipulated, problem behavior decreased for only two of four participants. It is unclear whether these mixed findings are a result of the number of reinforcer parameters that were manipulated (one versus multiple), because an individual was sensitive to a certain reinforcer parameter, or if appropriate values of the reinforcer were identified. As mentioned in Kunnavatana et al. [17], future research should investigate if multiple parameters to which someone is less sensitive can be combined to overcome a single most-sensitive parameter. Additionally, given that all four participants had different sensitivities to reinforcer characteristics, future research should conduct these assessments on a larger scale to explore the utility of these assessments in distinguishing different subtypes of ASD. 

Although the procedures employed in the current study resulted in many positive outcomes, they are not without limitations. In the current study, both conditions of FCT were effective. Therefore, lengthy assessments may not be necessary. It is possible that, at least for some individuals, an FCT without an extinction procedure in which the consequences are equal or the manipulated parameter(s) are chosen based off factors other than sensitivity assessments (i.e., resources available, context), would produce similar therapeutic effects. Thus, future research may consider comparing sensitivity assessment-informed contingencies to an uninformed condition (e.g., sub-optimal contingencies) to determine if the inclusion of the relative parameter sensitivity assessments is necessary. Related to this, it is important to note that differential reinforcement was in effect during the baseline. That is, reinforcers were delivered for both problem behavior and communication responses, with reinforcement favoring problem behavior. This deviates from the baseline arrangements discussed in previous FCT research. For example, in Kunnavatana et al. [17], problem behavior and the FCR resulted in equal reinforcers in the baseline. In other studies, baseline included no programmed consequence for the FCR [4]. While each arrangement has certain benefits and limitations, we argue that the unequal contingencies employed in the baseline condition of the current study likely mimic the natural environment that shaped problem behavior in the first place and strengthen the argument that FCT without extinction and parameter manipulations is effective in reducing problem behavior. Future research may consider employing two baseline conditions (one with equated reinforcing consequences for problem behavior and the FCR and one with no consequence for the FCR) that serve as a comparison to the effective treatment, similar to the baseline arrangement described in Briggs et al. [11]. 

Additionally, we want to recognize the limited clinical utility of the current study. It would be difficult to make the case that every assessment included in this study is necessary for all clients with problem behavior. Given participants were recruited through flyers posted in the community and the severity of problem behavior was not an inclusion criterion, extinction may have been appropriate and feasible for some of the participants in this study. Specifically, Dominick’s pulling and Maggie’s screaming were neither risky or unsafe behavior. Future research should consider evaluating an abbreviated version of our assessments and include participants with more severe problem behavior. Lastly, the FCT conditions evaluated in the current study are not intended to serve as terminal interventions. Ideally, we would have implemented schedule thinning following the reduction in problem behavior. Future research should evaluate schedule thinning following FCT with and without extinction and determine whether resurgence is more or less likely to occur when FCT includes parameter manipulations versus extinction. 

In conclusion, this study demonstrated that FCT without extinction, informed by the results of relative parameter sensitivity assessments, was effective in reducing problem behavior and increasing functional communication for four participants. The inclusion of the magnitude and delay value sensitivity assessments allowed for precise treatment programming and should be considered in future studies evaluating DRA without extinction interventions. Although further research is necessary, the manipulation of reinforcer parameters is an important area for continued investigation as they allow for the treatment of problem behavior through relatively nonobtrusive and reinforcement-based contingencies.

## Figures and Tables

**Figure 1 behavsci-14-00546-f001:**
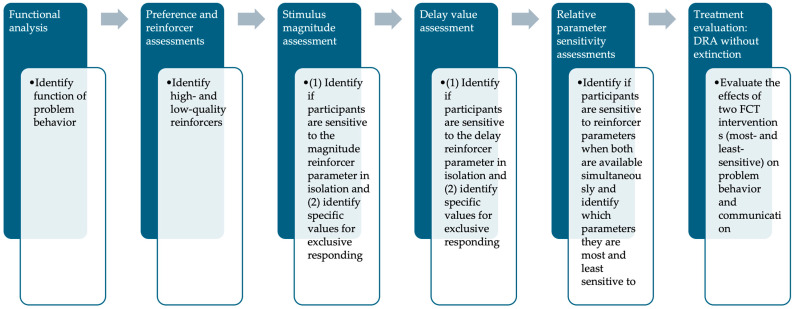
Experimental Operation Flow Chart.

**Figure 2 behavsci-14-00546-f002:**
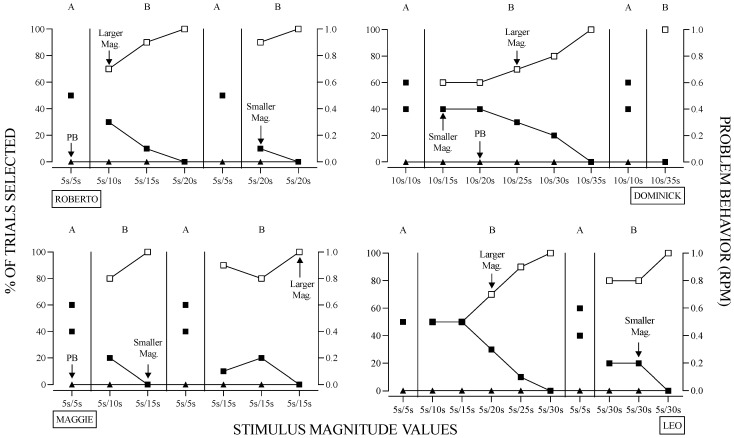
Results of the Stimulus Magnitude Assessment. *Note.* The X-axis contains the magnitude values (i.e., duration of access) concurrently available during the 10-trial session. All reinforcers were delivered immediately.

**Figure 3 behavsci-14-00546-f003:**
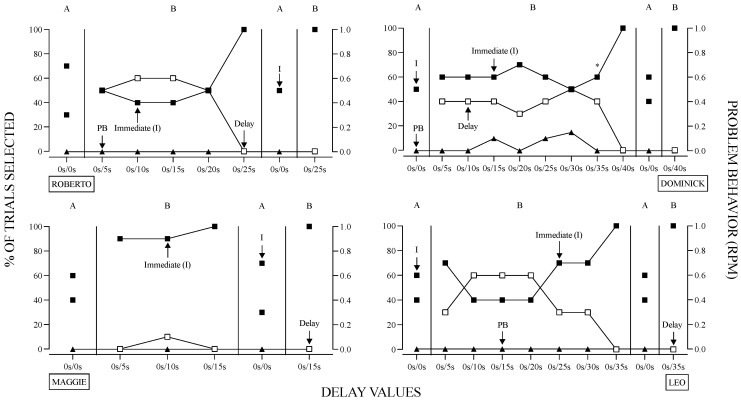
Results of the Delay Value Preference Assessment. *Note.* The X-axis displays the magnitude of the delay (i.e., latency to access) concurrently available during the 10-trial session. The magnitude value (20, 35, 15, and 30 s for Roberto, Dominick, Maggie, and Leo, respectively) remained constant. Asterisk denotes the introduction of a rule with Dominick.

**Figure 4 behavsci-14-00546-f004:**
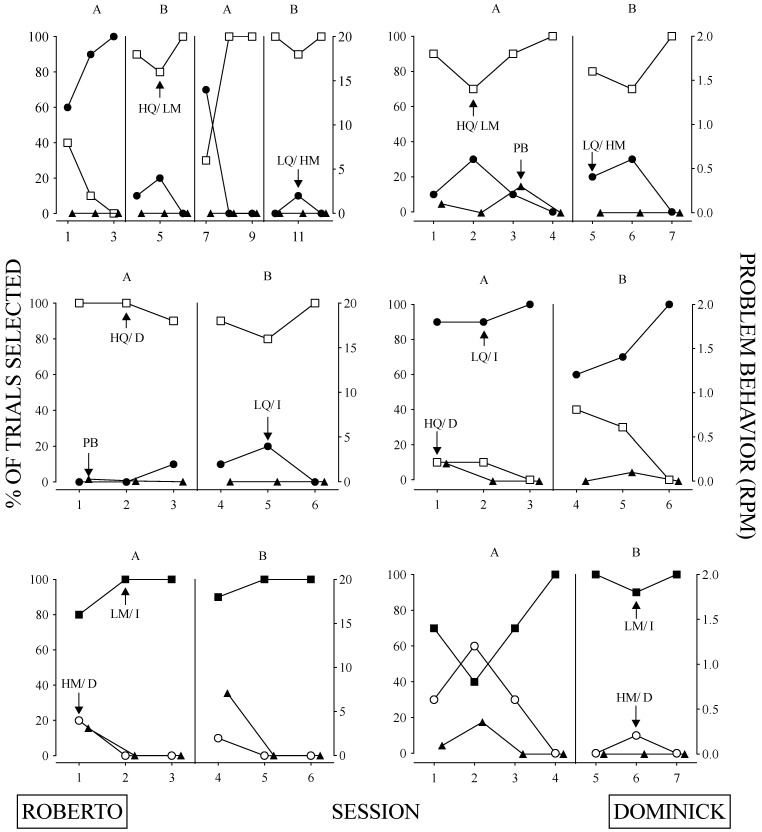
Results from the Relative Parameter Sensitivity Assessments for Roberto and Dominick. *Note*. Top panel depicts quality versus magnitude, middle panel quality versus immediacy, and bottom panel magnitude versus immediacy. The right Y-axis shows problem behavior (i.e., closed triangles).

**Figure 5 behavsci-14-00546-f005:**
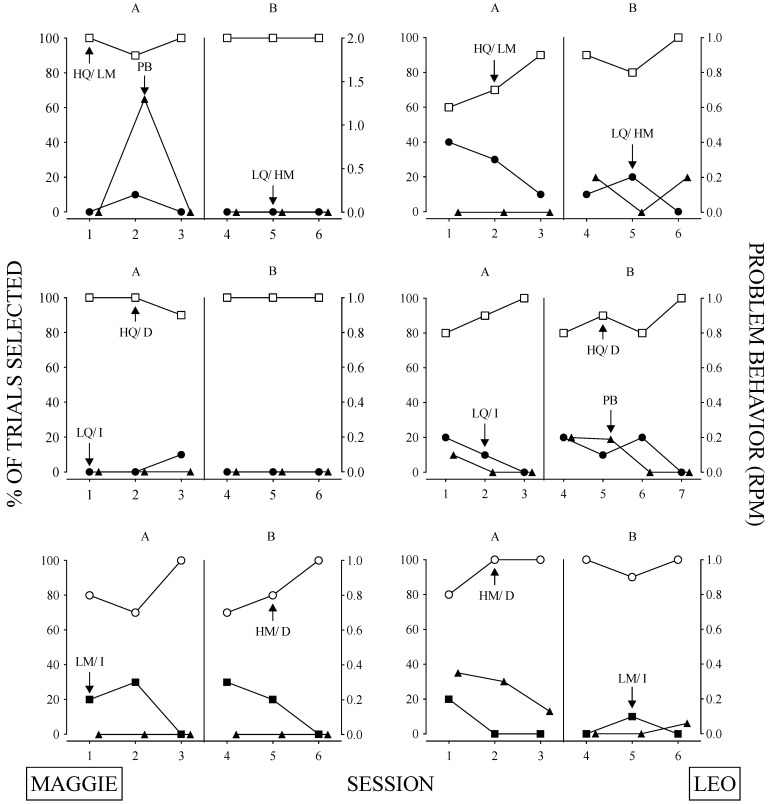
Results from the Relative Parameter Sensitivity Assessments for Maggie and Leo. *Note*. Top panel depicts quality versus magnitude, middle panel quality versus immediacy, and bottom panel magnitude versus immediacy. The right Y-axis shows problem behavior (i.e., closed triangles).

**Figure 6 behavsci-14-00546-f006:**
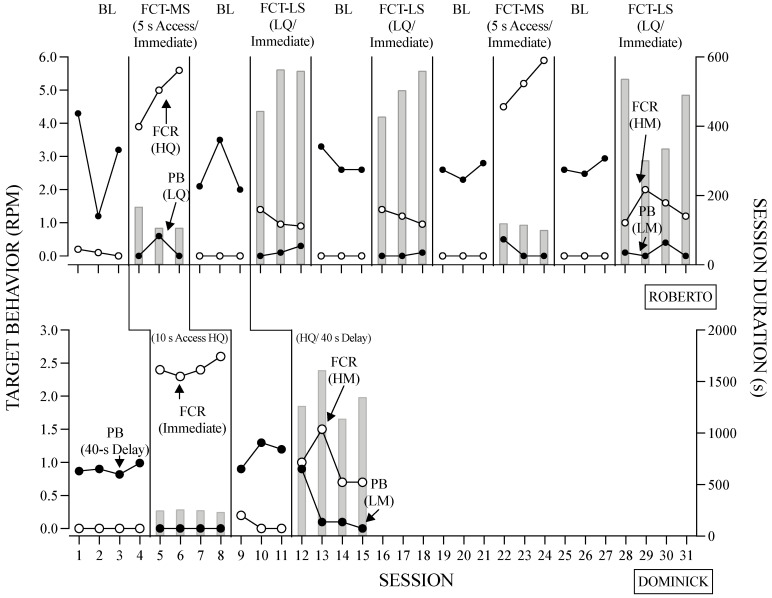
Results of the Treatment Evaluation for Roberto and Dominick. *Note*. The right Y-axis shows session duration (i.e., gray bars).

**Figure 7 behavsci-14-00546-f007:**
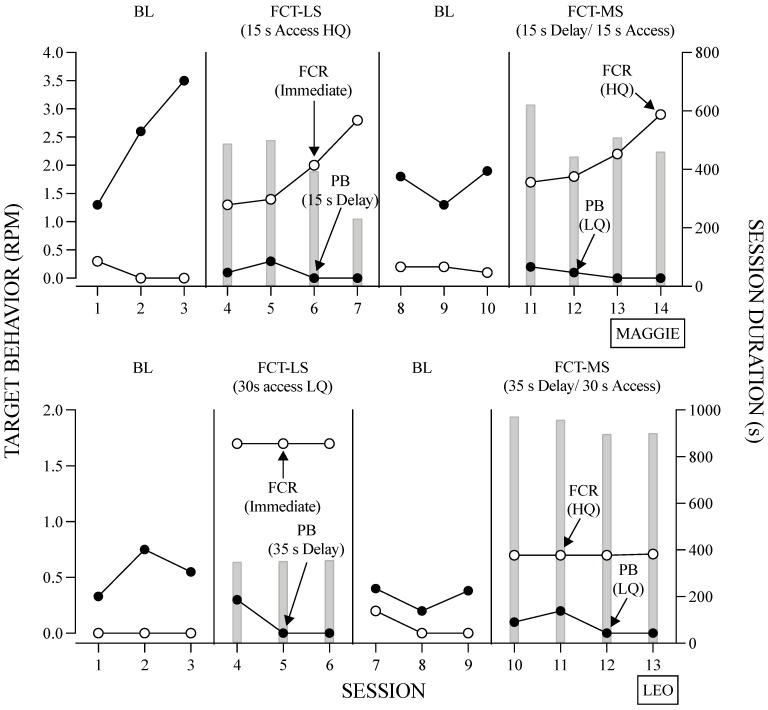
Results of the Treatment Evaluation for Maggie and Leo. *Note*. The right Y-axis shows session duration (i.e., gray bars).

**Table 1 behavsci-14-00546-t001:** Participant Information.

Participant	Age (yrs)	PB	Function of PB	FCR	Higher-Quality Reinforcer	Lower-Quality Reinforcer
Roberto	4	Aggression (hitting, kicking)	Tangible	Vocal verbal: “My turn please”	iPad^®^	Balloon/Ball
Dominick	2	Pulling clothing	Tangible	Picture card exchange: “I want toys”	iPad^®^/Pin art toy	Letters toy/Guitar
Maggie	2	Screaming	Tangible	Vocal verbal: “My turn please”	Bubbles	Guitar
Leo	10	Biting	Escape	Picture card exchange: “BREAK”	Break + iPad^®^	Break

*Note.* PB = problem behavior.

**Table 2 behavsci-14-00546-t002:** Participant’s Relative Parameter Sensitivity Assessment Contingencies.

Participants	Quality vs. Magnitude	Quality vs. Immediacy	Magnitude vs. Immediacy
Roberto	iPad^®^ (5 s mag) vs. Balloon (20 s mag); 0 s delay for both	iPad^®^ (25 s delay) vs. Balloon (0 s delay); 20 s mag for both	5 s mag (0 s delay) vs. 20 s mag (25 s delay); iPad^®^ for both
Dominick	iPad^®^ (10 s mag) vs. Letters toy (35 s mag); 0 s delay for both	iPad^®^ (40 s delay) vs. Letters toy (0-delay); 35 s mag for both	10 s mag (0 s delay) vs. 35 s mag (40 s delay); iPad^®^ for both
Maggie	Bubbles (5 s mag) vs. Ball (15 s mag); 0 s delay for both	Bubbles (15 s delay) vs. Ball (0 s delay); 15 s mag for both	5 s mag (0 s delay) vs. 15 s mag (15 s delay); Bubbles for both
Leo	Break + iPad^®^ (5 s mag) vs. iPad^®^ (30 s mag); 0 s delay for both	Break + iPad^®^ (35 s delay) vs. iPad^®^ (0 s delay); 30 s mag for both	5 s mag (0 s delay) vs. 30 s mag (35 s delay); Break + iPad^®^ for both

*Note.* Mag = magnitude.

**Table 3 behavsci-14-00546-t003:** Participant’s Treatment Evaluation Contingencies.

Participant	Reinforcer Parameters	Conditions	PB Contingency	FCR Contingency
Roberto	MS = Quality LS = Magnitude	BL	HQ (iPad^®^)/HM (20 s)	LQ (ball)/LM (5 s)
		FCT-LS	LQ (ball)/**LM (5 s)**	LQ (ball)/**HM (20 s)**
		FCT-MS	**LQ (ball)**/LM (5 s)	**HQ (iPad^®^)**/LM (5 s)
Dominick	MS = Immediacy LS = Magnitude	BL	HM (35 s)/I (0 s)	LM (10 s)/D (40 s)
		FCT-LS	D (40 s)/**LM (10 s)**	D (40 s)/**HM (35 s)**
		FCT-MS	**D (40 s)**/LM (10 s)	**I (0 s)**/LM (10 s)
Maggie	MS = Quality LS = Immediacy	BL	HQ (bubbles)/I (0 s)	LQ (ball)/D (15 s)
		FCT-LS	**D (15 s)**/HQ (bubbles)	**I (0 s)**/HQ (bubbles)
		FCT-MS	D (15 s)/**LQ (ball)**	D (15 s)/**HQ (bubbles)**
Leo	MS= Quality LS = Immediacy	BL	HQ (break + iPad^®^)/I (0 s)	LQ (break)/D (35 s)
		FCT-LS	LQ (break)/**D (35 s)**	LQ (break)/**I (0 s)**
		FCT-MS	**LQ (break)**/D (35 s)	**HQ (break + iPad^®^)**/D (35 s)

*Note.* BL = baseline; MS = most sensitive; LS = least sensitive; HQ = higher quality; LQ = lower quality; HM = high magnitude; LM = low magnitude; I = immediate; D = delay. Bold font indicates the reinforcer parameter that was manipulated.

## Data Availability

Data are contained within the article and Appendix A.

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
