# Peer review of "Differential Reinforcement without Extinction: An Assessment of Sensitivity to and Effects of Reinforcer Parameter Manipulations"

_behavsci, 2024, doi:10.3390/bs14070546_

Round 1

Reviewer 1 Report (Previous Reviewer 2)

Comments and Suggestions for Authors

This manuscript reads really well. The authors have done a nice job of addressing every concern that I had and should be commended for all the hard work in their revision. There are a few instances requiring minor editing, but I do not have any major concerns. 

Author Response

Thank you for reviewing our manuscript and providing a favorable response regarding the revisions. We are thrilled that you find the revised manuscript improved. 

Reviewer 2 Report (New Reviewer)

Comments and Suggestions for Authors

This study explored stimulus magnitude and delay value sensitivity assessments to identify optimal magnitude and delay values for each participant’s contingencies, and evaluated the efficacy and efficiency of two variations of FCT without extinction. It is somewhat innovative to expand on the basis of the existing research

But it has the following problems that need to be improved:

(1) The introduction of the experimental procedures is all text, not vivid enough, making it difficult for readers to clearly understand the specific operation of FCT. It is suggested to add the experimental operation flow chart.

(2) The study results found that the four subjects have different sensitivity to a reinforcer characteristics, which is an interesting finding. It is suggested to explore the use of sensitivity differences to a reinforcer to distinguish and even diagnose different types of ASD.

(3) The innovations relative to existing research should be highlighted in discussion.

(4)The main results should present in abstract.

Comments on the Quality of English Language

There were many professional terms lack of operational definition in Introduction and Method, leading to reading difficulties. It is suggested to add introduction of operational definitions.

Round 2

Reviewer 2 Report (New Reviewer)

Comments and Suggestions for Authors

The revised draft has addressed most of my questions, but there is still one issue that needs improvement.

In 2.4. General Procedure, the article showed: During the functional analysis (FA) we employed a multielement experimental design. A reversal design was employed during the stimulus magnitude assessment, the delay value assessment, and the relative parameter sensitivity assessments....

The description of these experimental processes is very general and not specific, and even after reading them, I still don't know how FCT is conducted? It is suggested the authors introduce the specific experimental process, such as what tasks the subjects need to do, under what circumstances they can obtain items as reinforcement. In this way, readers can repeat the experiment based on the experimental process description provided by the author to verify its reliability, validity, and reproducibility.

Author Response

Comment from reviewer:

The revised draft has addressed most of my questions, but there is still one issue that needs improvement.

In 2.4. General Procedure, the article showed: ”During the functional analysis (FA) we employed a multielement experimental design. A reversal design was employed during the stimulus magnitude assessment, the delay value assessment, and the relative parameter sensitivity assessments....”

The description of these experimental processes is very general and not specific, and even after reading them, I still don't know how FCT is conducted? It is suggested the authors introduce the specific experimental process, such as what tasks the subjects need to do, under what circumstances they can obtain items as reinforcement. In this way, readers can repeat the experiment based on the experimental process description provided by the author to verify its reliability, validity, and reproducibility.

Response from author: Thank you for this comment. We agree that this specific section (2.4 General Procedure) is very general. The detailed description of each experimental procedure is provided in their individual sections. To aid the readers in knowing that this information is forthcoming in the manuscript, I have added the following verbiage from this section:

"Figure 1 displays the experimental operations flow chart which outlines each step in the experimental process and lists the goal for each experimental operation. Then, detailed descriptions of each experimental operation are provided in their respective sections (e.g., section 2.10-2.13 describe the FCT evaluation)."

In these sections, I do believe that the procedures are detailed enough for replication. For instance, in 2.10 it states, "At the start of each session, the researcher presented the specified evocative situation to the participant. This included removing access to the tangible reinforcer but keeping them in sight for Roberto, Maggie, and Dominick and prompting Leo to read from a book. Consequences comprised of differing parameters based on the results of the sensitivity assessments were provided for problem behavior and FCRs on a fixed ratio (FR) 1 schedule." Then, in sections 2.12 and 2.13, FCT-MS and FCT-LS, we provide examples of the consequences for both target behaviors. Additionally, the specific consequences for each participant are provided in Table 3. 

I do hope that you agree this edit is sufficient, but if you do not, I am happy to move information around so that more detail is provided earlier on. 

This manuscript is a resubmission of an earlier submission. The following is a list of the peer review reports and author responses from that submission.

Round 1

Reviewer 1 Report

Comments and Suggestions for Authors

1. Some structures in the article need to be reorganized. For example, in the abstract, it should highlight the innovations of the work presented in this paper. The conclusion section should be included at the end of the paper.

2. The methods section lacks relevant models, equations, pseudocode, and other elements related to reinforcement learning techniques.

3. The paper is lacking a related work section, and the experiments lack comparisons with existing works.

4. The organization of the methods section is somewhat chaotic, and it is not clear which parts represent the techniques proposed in this paper and which parts are existing techniques. It is important to distinguish them from related work.

5. The issues addressed by this paper and its novelty should be clearly defined.

6. It is recommended to provide an overall framework diagram in the methods section.

7. The citation format of the references needs to be standardized.

Reviewer 2 Report

Comments and Suggestions for Authors

Thank you for the opportunity to review the manuscript titled “Differential Reinforcement without Extinction: An Assessment of Sensitivity to and Effects of Reinforcer Parameter Manipulations”. I reviewed this paper with some excitement as I too have conducted research evaluating different parameters of reinforcement. In this study, preference for reinforcement parameters were assessed in isolation and combination, which informed the development of FCT treatments that were compared by problem behavior, communication responses, and session length. I applaud the authors for their attempt at a rigorous assessment process that focused on experimental manipulations not often considered in clinical or research work. Despite the strong effort, I have numerous questions and concerns that preclude me from recommending acceptance. Two major issues that were difficult to overlook were the lack of description in the Methods section that would allow for replication and the inability to communicate the importance of conducting such procedures in practice or research. As you will see below, I have outlined  both comments and questions that contributed to my decision. There are a number of issues that could be addressed fairly easily (“Minor Consideration”, which often involve formatting, spelling, or simple errors) and issues would need to be addressed if to be considered for publication (“Major Considerations”, which may be difficult or impossible to address). If this manuscript is considered for revision, I hope that comments and questions are helpful.

Editing:

There were many spelling, grammatical, and formatting errors that could have been prevented by reading through the manuscript prior to submission. Below is just an example of those that I noted as I reviewed the manuscript:

·         Add an “s” to “procedure” or to “provide”.

·         Add “was” between “DRA” and “participants”.

·         Delete “as” in line 31.

·         Use “a” participant’s or “participants’” in line 68.

·         Individualized is misspelled on p 3.

·         Add hyphen after “high” on p 4.

·         No need to call them “handheld devices”. Just call them cell phones.

·         For data “collection” purposes, not data “analysis” purposes.

·         “measured” is misspelled on p 6.

·         Add “s” following “contain” on p 8.

·         Set off lists with a colon rather than a comma. There are several places in the manuscript where this should be addressed.

·         There are sentences where commas are needed.

·         Roman numerals are used in the results when referring to the figures, but the figures are not listed by roman numerals.

Introduction:

Minor considerations:

·         The first sentence is very long and could be broken up.

·         When reinforcement is withheld from a behavior that previously resulted in reinforcement that is extinction. Please make that more explicit in describing the procedure.

Methods:

Minor considerations:

·         Specify “vocal” communication for the participants.

·         Indicate that the mode (and language) chosen was based on parent preference rather than just “input”. Determination of how FCR modalities are chosen is important.

·         Report the session length(s).

·         There are several references that appear to be missing from the reference list. For example: Wolfe et al. (2018); Vessells et al. (2018).

·         Calling a reinforcer high-quality or low-quality is problematic. I realize that there is precedent (e.g., Kunnavatana et al., 2018), but the comparisons are relative and not absolute. I suggest using “higher-quality” and “lower-quality”. This should be the case for descriptions of magnitude as well.

·         I think that the characterization for quality of reinforcer for escape-maintained behavior (Leo) is problematic. The enriched break combines two reinforcers (positive and negative), and the “low-quality” reinforcer is singular (negative). For that case, consider an alternative description (e.g., enriched versus unenriched break or combined reinforcers versus single reinforcer) as quality does not seem to be a good description as these reinforcers are not varying on the same dimension.

·         The basis for treatment procedures without extinction is largely due to feasibility issues (e.g., too risk or unsafe, poor fidelity); however, for two of the children included in this study (Dominick and Maggie), neither appears to exhibit risky or unsafe behavior and both are very young. I’m curious why these two would be good candidates for extinction-less procedures. It would also be helpful to understand how or why these children were recruited for participation.

·         It appears that Dominick’s starting point for the magnitude assessment was 10s while all others were 5s. Any particular reason for this?

·         The purpose of the mand training and its procedures are confusing. Why was mand training necessary? If using a most-to-least prompting sequence, was extinction used? I assume that mand training sessions were also 10 trials.

·         Did an FCR and PB occur concomitantly during a delay trial? If so, how was that handled?

·         It sounds like a potential punishment procedure (i.e., resetting the delay) was used?

·         Clarify that the research team conducted all procedures and not the caregivers.

Major considerations:

·         The baseline is not described. It appears that the FCT evaluation served as the baseline, but this was not made explicit.

·         If the BL for the FCT evaluation entailed differential reinforcement for PB (i.e., PB resulted in favorable outcomes and FCR resulted in less favorable or aversive outcomes), this seems to be a poor choice for a baseline. In addition to deviating from traditional FCT (which should serve as an adequate baseline), it seems to create a strawman comparison that is devoid of scientific value. This is actually a comparison of three FCT without extinction treatments as the BL is also FCT and without extinction.

Results:

Minor considerations:

·         Add a note for the figures that defines the abbreviations used.

·         I think that the procedural modification for Dominick that occurred during the delay value assessment should be described in the Methods section along with greater detail (e.g., how was this rule presented).

·         There is an error in the first sentence describing Maggie’s results on p 10.

·         The nice thing about having data on problem behavior is that it adds another potential measure of sensitivity to the parameters. Consider reframing sensitivity based on both selection response and problem behavior. For example, Leo may find the quality of the reinforcer the most preferred dimension, but his problem behavior appears to be consistently higher when the quality of the reinforcer is constant (i.e., HM/D vs LM/I). Thus, Leo’s problem behavior (and not response selection behavior) may be most sensitive to the magnitude or delay of the reinforcer.

·         Consider scaling all of Maggie’s secondary y-axes at the same level so that comparisons are easier.

·         Add participant name labels Figure 5.

Major considerations:

·         It is unclear to me why the initial session in the individual stimulus magnitude and delay value assessments is not considered the “A” phase. This is the same condition conducted in the “B” phase. Thus, I recommend the initial session be the A phase, followed by the stimulus magnitude/delay value assessment serve as the B phase, then reversal to the A and B phases (ABAB).

·         There is no description of the distinction between the A and B phases during the relative parameter sensitivity assessments. They appear to be the same two conditions evaluated in each phase and there is no mention in the Methods of the distinction.

·         I am surprised that for sessions (with 10 trials apiece) involving such a low magnitude of reinforcement (5 – 30 s) with an iPad that there was no problem behavior during the stimulus magnitude assessment. Especially since 3/4 children demonstrated a tangible function during the FA. Were data collected on problem behavior only while consuming the reinforcer? If so, were problem behaviors observed during the intertrial intervals, at the point of restricting the preferred item, or during the delay interval during the delay assessment? This should be described and if problem behavior was observed but data were not collected on it, it should be discussed as a limitation because of the impact on others who would choose to conduct similar assessments (i.e., they should be aware of the potential for problem behavior).

·         It seems to me that the use of RPM for the FCR during the FCT analysis is problematic. If each session consisted of 10 trials, it does not make much sense to compare the RPM of the FCR. This will be determined by the session length (which is not defined in the Methods) and would fluctuate based on the parameters being manipulated (e.g., a session consisting of trials where the FCR resulted in 10 s access would have very different rates than a session where access was 40 s). A better comparison of the robustness of the FCR should be based on the percentage of opportunities used.

·         The social validity data is not helpful without the actual questions. The description of the results does not provide detail on how the questions were worded or whether there was an option for comparison of the two treatment conditions (MS and LS).

Discussion:

Minor considerations:

·         Line 432 says that “…no problem behavior was observed during these assessments…”. This is not true.

Major considerations:

·         See my previous note regarding the use of RPM for FCR data. It is not true to say that one treatment produced an increase in FCRs when the data were based on disparate session lengths.

·         It also seems problematic to describe the FCT-MS condition as more efficient for 2/4 participants when the session time was structural and not contingent on child behavior. In other words, the session time was pre-determined for the two options.

·         A very important limitation is that the assessment process is very lengthy and may not be any more efficient than initiating FCT following the FA (we do not have comparative data so say either way). The length of the assessment (without comparison to the standard approach) should be described as a limitation.

·         The main finding of the treatment evaluation (i.e., the resulting data show that PB and communication inversely covary depending on the favorable reinforcement contingency) is not substantially beneficial. It is possible that FCT without consideration for preference of reinforcer dimension may have shown the same effects.

·         Given the focus on treatment without extinction, it may be helpful to consider whether avoiding extinction is ultimately what is best for the individuals be served? As interventions are implemented across environments it is likely there will be instances where the desired FCR will not be honored (i.e., placed on extinction), either intentionally or unintentionally. In these instances, it is not only possible but probable that problem behavior will resurface again (i.e., resurgence), which means that the only way to avoid problem behavior is to honor the FCR at all times. While I can appreciate caregivers may indicate a preference to avoid extinction, I do not think it is possible or probable to avoid it in all circumstances. Thus, I think some of the potential negative consequences of avoiding extinction should be addressed in this section.